# Temperature Effect on Stability of Clamped–Clamped Composite Annular Plate with Damages

**DOI:** 10.3390/ma14164559

**Published:** 2021-08-13

**Authors:** Dorota Pawlus

**Affiliations:** Faculty of Mechanical Engineering and Computer Science, University of Bielsko-Biala, 43-309 Bielsko-Biala, Poland; doro@ath.bielsko.pl

**Keywords:** composite laminate annular plate, thermal loading, stability, damages, finite difference method, finite element method

## Abstract

The paper presents the response of a three-layered annular plate with different damaged laminate facings to the action of the static or dynamic temperature field model. Various damages of laminate, composite facings change the plate structure reaction under the temperature fields. Obtained results indicate practical meaning of analyses in failure diagnostic process. The thermal sensitivity of two kinds of plate structures, undamaged and damaged, offers both new practical and scientific possibilities in evaluation of the plate behavior. The relations between macro-damage, i.e., the buckling of the plate structure and micro-damages of plate layers subjected to temperature gradient, are shown. The numerical solution is proposed as the most effective in examinations of the various transversally symmetrical and asymmetrical plate structures with a different rate of damages. The graphical distribution of changes in values of static and dynamic critical loads illustrate the process of structural damaging during its exploitation. They have practical importance in the evaluation of the structure capacity. The knowledge of the effect of laminate degradation process on plate buckling phenomenon located in thermal environment complements previous investigations and designates complex, multi-parameter problems as having scientifically new elements.

## 1. Introduction

The range of possible applications of the annular plates is extensive, for example, in the aerospace industry, mechanical and nuclear engineering, civil engineering, or miniature mechanical systems. Plates can be subjected to the various and complex properties of the surrounding environment. These include the temperature field parameters. The action of the temperature gradient in the plate radial direction can be the reason for the loss of plate stability. Then, stress-strain plate parameters are expressed by critical ones. Buckling phenomenon can be used to evaluate the state of the composite structure and the degree of existing defects. During the exploitation process, it is not easy to predict cracks of fibers and matrix in the laminate material. Cracks of fibers or matrices are a form of failure of fibrous composite. Furthermore, the loss of the plate stability is a form of global failure. However, the detailed observations of the composite plate buckling behavior can help to evaluate the changes which occur in structure. The process of propagation of damages, such as fiber or matrix cracks in laminas, changes the geometry of the structure from transversally symmetric to asymmetric and alerts its rigidity parameters. The original plate problem becomes a new thermo-mechanical one and formulates a new task requiring the solution. 

The main novelty of the presented problem is the evaluation of the fiber-reinforced composite plate behavior depending on the static and dynamic temperature field action. A numerically three-layered annular plate with foam core and laminated facings was examined. Models with various combinations of facings damages were built to analyze different degrees of plate structure degradation. Transversally symmetrically and asymmetrically located cracks of fibers and/or matrix of facings laminas have formed the plate structure and analyzed plate models. The problem was solved numerically using the finite element method and analytically and numerically with the use of approximation methods, such as the finite difference one, in order to compare results of quasi-isotropic plate models. Values of critical temperature differences, which characterize the plate critical thermal state are analyzed in detail. Results are presented graphically to show the nature of changes of both the degree of structure degradation and corresponding values of critical temperature differences. The nature of the changes is sequential and orderly. Hence, the presented approach to the problem and the presentation of results is of practical importance in the failure diagnostic process performed to evaluate the state of the structure. 

According to the author’s knowledge, the formulated problem and idea to observe the plate buckling geometry and to analyze critical values of temperature differences between plate edges in the evaluation process as the effect of lamina cracks on the plate response have not been sufficiently considered. The main aim of this contribution is to evaluate the stability response of composite layered plates subjected to thermal loading, whose laminate facings can be variously defected. The second aim of the investigations is to present proposal analytical and numerical procedures of problem analysis, which is based on the mathematical calculations of structure rigidity and the application of the finite element (FEM) plate model in buckling static or dynamic calculations connected with thermal loads. Furthermore, the third aim of analyses presented in the paper can be formulated. The analytical and numerical solution, which uses the finite difference method (FDM) for a selected group of three-layered plates with laminate undamaged and damaged facings treated as quasi-isotropic are presented as well as the results comparison between FEM and FDM quasi-isotropic plate models. 

The review of literature connected with the undertaken problem can be described within three subject areas: stability problem, structure degradation process, and temperature loads. Exemplary dynamic stability analyses of sandwich annular plates are presented in [1,2]. The problem of axisymmetric buckling of laminated composite circular and annular plates is presented in [3,4]. Author’s papers which focus on the dynamic stability problem of the mechanically loaded three-layered annular plate are referred to below [5,6]. The presented solution to this problem under analysis, based on the orthogonalization and finite difference approximation methods, is used in this paper. Stability analysis of the three-layered plate with laminate facings is considered in this paper [7]. 

The structure degradation process, an effect of micro cracks or failure damages, is examined in the presented selected papers for mechanically loaded plates. The quasi-isotropic composite circular plate under quasi-static lateral load and low-velocity impact tests is presented in [8]. The analysis was performed with the use of the non-linear approximation method and the large deflection plate theory. The results show that the low-velocity impact responses are close to the quasi-static behavior of the plate. The fiber damage image, along with the damage propagation from the center of plate towards to the edge, are presented for plates with varying thicknesses. The thin-walled sandwich rectangular plates with axially compressed composite faces are examined in [9]. The transverse full symmetry in a plate of sandwich structure, which is composed of two multi-layered fiber reinforced plastic (FRP) faces, is examined. 

The bifurcation instability presented for rectangular plates made of fibrous composite materials with reinforcement subjected to long-term damage is formulated and solved in [10]. The problem of matrix cracking and delamination in laminated composites is presented in [11]. A model for the prediction of the propagation process of transverse cracks in polymer matrix composite laminates is proposed. Various crack patterns are analyzed. The failure model for the simulation of change in laminated composite plates is presented in [12]. Plates are subjected to dynamic loading. Matrix cracking propagation is analyzed numerically. The mathematical formulation for the modelling of damage in laminated composite plates and shells is presented in [13,14]. The micromechanical model for predicting the impact damage of composite laminas is proposed in [15]. The model is based on the laminate microstructure and various failure such as matrix cracking, fiber breakage, and delamination. The non-destructive testing method to characterize the composite material local damage is presented in [16]. Furthermore, [17] presents the use of the finite element analysis procedure, which is developed to predict the initiation and propagation of damages in laminated composite plates. 

The author’s papers showing the buckling behavior of laminated composite annular plates with structure failure subjected to mechanical loading are presented in [18,19,20]. An attempt to create a full image of plate stability responses is shown in a graphic form similar to the one assumed in this paper. In addition, the use of the mathematical formulae is proposed in [21], which modifies the elements stiffness matrix of composite structure. 

Thermal or thermo-mechanical responses of plates are the subject of the numerous works. Some of them are mentioned in this part of the introduction. A chapter of book [22] is dedicated to stability problem analysis of thermally loaded general heterogeneous FG annular plates. The buckling of heated FGM annular plates on elastic foundation is presented in [23]. Thermal buckling analysis of transversally graded circular plate subjected to a central elastic foundation is presented in [24]. Paper [25] presents a solution for a moderately thick annular plate made of FGM composite. Furthermore, the effects of the uniform temperature rise and heat conduction across the plate thickness on critical buckling are considered. The non-axisymmetric buckling behavior of annular plates subjected to temperature rise is presented in [26]. The problem of the asymmetric buckling behavior of isotropic homogeneous annular plates on a partial elastic foundation under uniform temperature is investigated in [27]. The case of moderately thick annular FGM plates on partial elastic foundation under uniform temperature rise, including the temperature dependent material properties, which are graded across the plate thickness, is presented in [28]. The critical buckling temperature is examined for plates with a temperature-dependent elasticity modulus and a thermal expansion coefficient. The problem of the FG annular plate loaded with thermal shock is presented in [29]. The critical buckling and dynamic post-buckling responses of the FGM annular plates with initial geometric imperfections are considered in [30]. The effects of the loads and material parameters and imperfection rates on dynamic behaviors and values of critical temperatures are examined in detail. The computational analysis of the non-linear vibration and thermal post-buckling of a heated orthotropic annular plate is presented in [31]. The axisymmetric thermal buckling analysis of thin FG annular plates based on von Karman’s plate theory is undertaken in [32]. Vibration evaluations used in the optimization of elastic annular plates subjected to thermal loading and vibrations thermally induced by oscillating heat stream are presented in [33]. Free vibration analysis of FG annular plates with various thickness in the thermal environment supported on elastic foundation is investigated in [34].

Plates models subjected to thermal environment are the subjects of consideration in author’s works [35,36,37]. The solutions for three-layered annular plates which are differently supported and loaded with stationary temperature field are employed in [35]. The evaluation of behavior of three-layered annular plate made of steel facings and supported in slideably clamped edges subjected to time-dependent temperature field and two dynamic fields, mechanical and thermal, is presented in [36]. The analysis focused on the effect of viscoelastic core of a three-layered annular plate with homogeneous steel facings on responses due to mechanical and/or thermal loading is presented in [37]. Results show the dynamic stability of plate models supported with clamped edges when the plate is loaded only thermally and plates with slideably clamped edges for plates which are located in the thermal environment and mechanically loaded.

The literature review shows an insufficient range of investigations, which undertake the combined problem of buckling, as well as static and dynamic analysis and thermal loading performed for composite plates with micro-damaged laminate layers. Such an issue is the multi-parameter problem dependent on geometrical, material, and structural properties of the plate, connected with a support system and a somewhat loading state. Additionally, the procedure of plate modelling based on the approximation methods has influence on the accuracy of final results. Graphically presented results will show the character of changes of critical temperatures, which denotes ordered responses of damaged and thermally loaded plates. Such an observation initiates the research problem with new elements leading to diagnostic evaluation of composite plates in temperature fields. So, the analyzed problem is scientifically interesting and practically important. 

## 2. Problem Formulation

A three-layered annular plate with thin laminated facings and a thicker core was placed in a thermal environment. Temperature differences between the inner and outer plate edge created thermal gradients of temperature fields surrounding plate perimeters. The plate was subjected to a loss of stability. The critical state depended on various parameters which were also connected with the quality of laminated plate outer layers. Fiber or matrix cracks in laminas changed the structure properties and the ability of the plate yield to work in variable conditions. Then, the plate cross-section geometry can be transversally symmetrical or asymmetrical. Including both the symmetrical and asymmetrical forms of plate buckling and different cases of lamina damages, the evaluation of the temperature field effect on plate stability was analyzed for two environmental models: static and dynamic. Dynamically, the plate was loaded with temperature difference expressed by the following relation (see, the main notations presented in Table A1—part Appendix A of paper):Δ*T* = *at*(1)

Figure 1 shows a scheme of the clamped–clamped plate in a thermal environment with temperatures *T_i_* and *T_o_* in the area of plate hole and outer perimeter, respectively. The undertaken dynamic stability problem requires the adoption of a stability loss criterion. The criterion presented by Volmir in [38] was adopted. According to this criterion, the plate stability loss occurs at the moment when the speed of the point of maximum deflection reaches the first maximum value. Black dots shown in Figure (see, for example Figure 2) presented in the paper signify the moment of the plate dynamic stability loss.

Plate facings composed of four laminas with fibers were arranged according to the code [0/−45/45/90]. The configuration of laminas fulfilled the conditions of the quasi-isotropic composite. This allowed for a comparison of results between two plate models built using the finite difference method (FDM) and the finite element method (FEM). The two plate models are examined with isotropic or orthotropic thermal properties.

The temperature field was axisymmetrical and flat. There was no heat exchange between the plate surfaces. The heat flow, defined by the logarithmic distribution of temperature versus the plate radius (see, equation (20)), existed only in the radial direction of plate facings. Material constants did not depend on temperature. Two cases of temperature changes were examined with positive and negative thermal gradients, when the temperature *T_i_* in the plate whole was higher than in the outer surroundings (*T_i_* > *T_o_*, see Figure 1) and when the temperature values were opposite (*T_i_* < *T_o_*), respectively.

## 3. Problem Solution

The solution to the problem required the use of the numerical approximation methods. Two variants of the solution were proposed. The first of them was based on the orthogonalization and finite difference methods (FDM) plate model. The problem of the clamped–clamped classical three-layered plate was solved analytically and numerically. The solution was conducted for quasi-isotropic thermo-mechanical properties of plate layers was expressed by the engineering constants, such as Young’s modulus—*E*, Kirchhoff’s modulus—*G*, Poisson’s ratio—*ν*, mass density—*μ*, and linear expansion coefficient—*α* in thermally isotropic facings.

The second variant of the solution to the problem was based on the finite element method (FEM) plate model. The plate model was built using the ABAQUS system. FEM plate models made it possible to observe the static and dynamic stability behavior of each of the examined structures: a composite without any defects, treated as quasi-isotropic one, and a composite having fiber or matrix cracks in selected facing laminas. That includes plates whose thermal properties are isotropic or orthotropic.

Both used methods were approximate. In the case of the FDM method, the basic inaccuracies were linked to the essence of method. It was connected with the demand that continuous function must be substituted by the set of discrete values. The convergence of the method with the grid concentration is the fundamental expectation (see, exemplary results presented in Table 1 and Table 2). The accuracy of the FEM method, connected with the process of the “idealization” of system geometry and the disruptions of the strains continuity, depended on the form of shape function describing the strain state of element and the degree of division into elements (see, point 3.3 of the paper).

Both presented methods of the solution to the problem based on the thermo-mechanical relations for laminate facings. The classical lamination theory was used [21,39,40]. The composite degradation of plate structure was expressed by the calculated and suitably modified plate rigidity parameters [21].

### 3.1. Thermo-Mechanical Model of Fibrous Composite

The physical relation for linear elastic theory is expressed by:(2)$$\sigma_{i}=Q_{ij}\left(\varepsilon_{j}-\alpha_{j}\Delta T\right)$$
where:

*α*—thermal expansion,

*Q*—stiffness matrix.

Using the classical lamination theory [21,39,40] the resultant forces and moments are expressed as follows:(3)$$\left\{N\right\}=\left[A\right]\left\{\varepsilon \right\}+\left[B\right]\left\{k\right\}-\left\{N^{T}\right\}$$
(4)$$\left\{M\right\}=\left[B\right]\left\{\varepsilon \right\}+\left[D\right]\left\{k\right\}-\left\{M^{T}\right\}$$
where:

$\left\{N\right\},\left\{M\right\}$—vectors of resultant forces and moments, respectively,

$\left\{N^{T}\right\},\left\{M^{T}\right\}$—vectors of resultant thermal forces and moments, respectively,

$\left[A\right],\left[B\right],\left[D\right]$—matrixes of extensional coupling and bending stiffness, respectively,

$\left\{\varepsilon \right\},\left\{k\right\}$—vectors of strains and curvatures of the middle surface, respectively.

The elements of stiffness matrixes $\left[A\right],\left[B\right],\left[D\right]$ of laminate facings are expressed by the following relations:(5)$$A_{ij}=\sum_{k=1}^{N}{\left({\bar{Q}}_{ij}\right)}_{k}\left(z_{k}-z_{k-1}\right)$$
(6)$$B_{ij}=\frac{1}{2}\sum_{k=1}^{N}{\left({\bar{Q}}_{ij}\right)}_{k}\left(z_{k}^{2}-z_{k-1}^{2}\right)$$
(7)$$D_{ij}=\frac{1}{3}\sum_{k=1}^{N}{\left({\bar{Q}}_{ij}\right)}_{k}\left(z_{k}^{3}-z_{k-1}^{3}\right)$$
where ${\bar{Q}}_{ij}$—transformed reduced stiffness matrix of lamina, *N*—number of layers, *z_k_* and *z_k−_*_1_—coordinates in cross-section laminate of the outer surfaces of layer numbered as *k* and *k −* 1 with thickness equal to *t_k_*, respectively.

Thermal forces and moments are described by:(8)$$\left\{N^{T}\right\}=\Delta T\sum_{k=1}^{N}{\left[\bar{Q}\right]}_{k}{\left\{\bar{\alpha }\right\}}_{k}t_{k}$$
(9)$$\left\{M^{T}\right\}=\Delta T\frac{1}{2}\sum_{k=1}^{N}{\left[\bar{Q}\right]}_{k}{\left\{\bar{\alpha }\right\}}_{k}\left(z_{k}^{2}-z_{k-1}^{2}\right)$$
where

$\left\{\bar{\alpha }\right\}$—the vector of the apparent thermal expansion.

The apparent *α_x_*, *α_y_*, and *α_xy_* rates of thermal expansion in laminate axis (*x*,*y*) were calculated for laminas treated as thermally orthotropic. The apparent rates of thermal expansion were obtained using the plus and minus transformation of orthotropic thermal expansion rates *α*_1_, *α*_2_ in the main material axis 1, 2.

The elastic, engineering constants *E*, *G*, *ν* for configuration of quasi-isotropic composite are expressed by the following formulae [21]:(10)$$E=2\frac{A_{66}}{t_{l}}\left(1+\frac{A_{12}}{A_{11}}\right),\;G=\frac{A_{66}}{t_{l}},\nu =\frac{A_{12}}{A_{11}}$$
where *A*_11_, *A*_12_, *A*_66_—extensional stiffness *A_ij_* (*i*, *j* = 1, 2, 6).

### 3.2. Composite Degradation Model

Fiber or matrix cracks in the plate facings changed the mechanical properties of the laminate and the rigidity of the plate structure. The accepted model of the composite degradation was based on the theory of correction parameter method, presented in [21]. The mathematical essence of this method was based on the modification of the stiffness matrix, whose form for undamaged lamina is expressed by the following elements:(11)$$Q_{11}=\frac{E_{1}}{\left(1-\nu_{12}\nu_{21}\right)}{,\;Q}_{22}=\frac{E_{2}}{\left(1-\nu_{12}\nu_{21}\right)}{,\;Q}_{12}=\frac{E_{1}\nu_{21}}{\left(1-\nu_{12}\nu_{21}\right)}{,\;Q}_{21}=\frac{E_{2}\nu_{12}}{\left(1-\nu_{12}\nu_{21}\right)},\;Q_{66}=G_{12}$$

It is assumed that a matrix crack eliminates rigidity in the direction transverse to the fibers. It is expressed by the correction parameter *η*. For the lamina with a matrix crack, elements *Q*_11_, *Q*_12_, *Q*_22_ take on the following new values:*Q*_11_*= η Q*_11_, *Q*_12_*= Q*_22_*=* 0.(12)

When a fiber crack occurs, the stiffness matrix modification is limited to a replacement of elements *Q*_11_ by *Q*_22_ [21]:*Q*_11_*= Q*_22_(13)

The analyzed problem of plates with quasi-isotropic composite facings was solved analytically and numerically using the orthogonalization method and the finite difference method (FDM), and only numerically using the finite element method (FEM). The FEM makes it possible to thoroughly observe the plates with quasi-isotropic composite facings and composite facings with damages. The examinations were conducted for plate models with facings being damaged in the form of fiber or matrix cracks of a single lamina or all laminas.

### 3.3. Plate Model Built Using the Finite Element Method

The calculations were carried out using the ABAQUS system at the Academic Computer Center CYFRONET-CRACOW (KBN/SGI_ORIGIN_2000/Płódzka/030/1999). The full annulus plate model was built of shell elements and solid elements creating a mesh of the facing and the solid, respectively. The outer surfaces of facings and core mesh elements are tied using a program option expressed as surface contact interaction. The numbers, which characterize the FEM plate model are as follows: number of elements—3042 including number of internal elements generated for contact—2232, number of nodes—10044 including number of internal nodes generated for contact—4464, total number of variables in the model—30132.

The structural stiffness of plate facings was expressed by the elements of matrixes *A_ij_*, *B_ij_*, *D_ij_* (5 – 7), which were calculated separately and introduced in the shell option of the ABAQUS system. The stiffness elements *A_ij_*, *B_ij_*, *D_ij_,* were modified separately for each facing lamina according to the analyzed case of facing failure. The relations (11) – (13) of the composite degradation were used. The Buckle and Dynamic options of the program were used for the examination of the static and dynamic stability problem. The temperature influence was taken into account by the expression of vectors of thermal forces and moments $\left\{N^{th}\right\}$ [41]. The formula for thermal forces $\left\{N^{th}\right\}$, which change linearly with temperature and include thermal expression *α*, which is not a function of temperature is presented as follows:(14)$$\left\{N^{th}\right\}=\alpha \left(\theta -\theta_{I}\right)\left\{\bar{F}\right\}$$
where

$\left\{\bar{F}\right\}$—vector of stresses caused by a fully constrained unit temperature rise [41], whose elements were calculated using the geometry and material expressions of relations (8) and (9),

*θ_I_*—stress-free initial temperature,

*θ*—the temperature.

The value of thermal expression *α* was inputted separately as the scaling modulus for vectors $\left\{\bar{F}\right\}$ in the case of thermal isotropy of facing material. Including thermal orthotropy of laminate material, the apparent thermal expansions were taken into account, calculating the vectors $\left\{\bar{F}\right\}$. Then, the required scaling modulus is equal to 1.

### 3.4. Plate Model Built with Finite Difference Method

The variant of the solution to the problem refers to the procedure used in a numerical analysis of mechanically loaded three-layered annular plates. The problem has been described in works [5,6,42] in detail. The proposed solution was based on the classical theory of sandwich plates. The broken line hypothesis and the division of stresses into normal loading for the plate facings and shear load for the core were accepted. The main elements of the solution process were based on the description of the dynamic equilibrium equations and the linear physical relations, the expression of the geometry non-linear equations for facings using the Kármán’s equations, the introduction of the stress function, the determination of the supported conditions, and initial boundary conditions. After algebraic operations, a basic differential equation of plate dynamic deflections was obtained:(15)$$\begin{array}{c} k_{1}w_{d}{}_{,rrrr}+\frac{2k_{1}}{r}w_{d}{}_{,rrr}-\frac{k_{1}}{r^{2}}w_{d}{}_{,rr}+\frac{k_{1}}{r^{3}}w_{d}{}_{,r}+\frac{k_{1}}{r^{4}}w_{d}{}_{,\theta \theta \theta \theta }+\frac{2(k_{1}+k_{2})}{r^{4}}w_{d}{}_{,\theta \theta }+\\ +\frac{2k_{2}}{r^{2}}w_{d}{}_{,rr\theta \theta }-\frac{2k_{2}}{r^{3}}w_{d}{}_{,r\theta \theta }+\frac{2k_{2}}{r^{2}}w_{d}{}_{,rr\theta \theta }-\frac{2k_{2}}{r^{3}}w_{d}{}_{,r\theta \theta }+\\ -G_{2}\frac{H'}{h_{2}}\frac{1}{r}\left(\gamma_{,\theta }+\delta +r\delta_{,r}+H^{'}\frac{1}{r}w_{d}{}_{,\theta \theta }+H'w_{d}{}_{,r}+rH'w_{d}{}_{,rr}\right)=\\ \frac{2h'}{r}\left(\frac{2}{r^{2}}\Phi_{,\theta }w_{,r\theta }-\;\frac{2}{r}\Phi_{,\theta r}w_{,\theta r}+\frac{2}{r^{2}}w_{,\theta }\Phi_{,\theta r}-\frac{2}{r^{3}}\Phi_{,\theta }w_{,\theta }+w_{,r}\Phi_{,rr}+\Phi_{,r}w_{,rr}+\right.\\ \left.+\frac{1}{r}\Phi_{,rr}w_{,\theta \theta }\right){-\mathrm{Mw}}_{d_{,\mathrm{tt}}} \end{array}$$
where *H’ = h’ + h*_2_, *k*_1_
*= 2D*, *k*_2_ = 4*D_rθ_ + νk*_1_, $D=\frac{Eh{'}^{{}^{3}}}{12\left(1-\nu^{2}\right)}$, $D_{r\theta }=\frac{Gh{'}^{3}}{12}$—flexural rigidity of the outer layers, *M =* 2*h’μ + h*_2_*μ*_2_.

In order to obtain the basic differential system of equations of plate deflections, we employed shape functions of additional plate deflections, preliminary deflections, and stress function. The plate model had a preliminary deflection expressed by the function fulfilling the conditions of the clamped–clamped edges. Some of the dimensionless quantities (16) and shape functions of additional deflection (17), preliminary deflection (18) and stress (19) are as follows:(16)$$\;\zeta_{1}=\frac{w_{d}}{h},\;\zeta_{o}=\frac{w_{o}}{h},\;F=\frac{\Phi }{Eh^{2}},\;\rho =\frac{r}{r_{o}}{,\;t}^{*}=t\frac{a}{\Delta T_{f}},$$
(17)$$\zeta_{1}\left(\rho ,\theta ,t\right)=X_{1}(\rho ,t)\cos(m\theta )$$
(18)$$\zeta_{o}(\rho ,\theta )=\xi_{1}\eta_{o}\left(\rho \right)+\xi_{2}\eta_{o}(\rho )\cos(m\theta )$$
(19)$$F(\rho ,\theta ,t)=F_{a}(\rho ,t)+F_{b}(\rho ,t)\cos(m\theta )+F_{c}(\rho ,t)\cos(2m\theta )$$
where *ξ*_1_, *ξ*_2_—calibrating numbers, *η_o_(ρ) = ρ*^4^
*+ A*_1_*ρ*^2^
*+ A*_2_*ρ*^2^ln*ρ +*
*A*_3_ln*ρ +*
*A*_4_, *A_i_*—quantities fulfilling the conditions of the clamped edges by the function *η_o_*(*ρ*), *i =* 1, 2, 3, *r_o_*—outer radius of the annular plate.

The temperature distribution is a function of the plate radius and it is expressed by a logarithmic equation, according to theory presented in [43].
(20)$$T_{N}=T_{o}+\frac{T_{i}-T_{o}}{\ln\rho_{i}}\ln\rho$$
where *T_i_*, *T_o_—*temperatures of the inner and outer plate perimeters, (see, Figure 1).

The conditions for the thermally loaded plate with clamped–clamped edges are expressed by the Equations (22) and (23) established at discrete points 0 and *N* + 1, which are the points of the plate support. Equations (22) and (23) were obtained from Equation (21), which is expressed by means of the relations of the Hooke’s law in the plane stress state performed for normal forces with thermal elements in plate facings and after the elimination of the radial *r* and circumferential *θ* strains, and the acceptance of stress function *Φ*:(21)$$F_{'\rho \rho }-\frac{\nu }{\rho }\left(F_{'\rho }+\frac{1}{\rho }F_{'\theta \theta }\right)=-S\cdot T_{N}$$
where $S=\frac{r_{o}^{2}}{h^{2}}\alpha$.

Approximating the derivatives at points 0 and *N* + 1 with the use of the FDM differences in front and back the following relations were established:(22)$$\mathrm{for}\;\rho =\rho_{i}y_{o}=\frac{b\rho_{i}}{\rho_{i}+b\nu }\left(\frac{y_{1}}{b}+S(T_{o}+at)\right)$$
(23)$$\mathrm{for}\;\rho =\rho_{o}y_{N+1}=\frac{b}{1-b\nu }\left(\frac{y_{N}}{b}-ST_{o}\right)$$
where:$y_{o},y_{1},y_{N},y_{N+1}$—elements of vector $Y=F_{'\rho }$ of stress function at discrete points 0, 1, *N*, *N* + 1.

The solution process required a lot of algebraic operations and the use of the orthogonalization method. Then, by the approximation of the derivatives with respect to *ρ* by the central differences at discrete points, the following system of equations for the three-layered annular plate in the thermal environment was obtained:(24)$$PU+Q=K\cdot \ddot{U}$$
(25)$$M_{Y}Y=Q_{Y}-\rho \cdot S\cdot T_{N'\rho }$$
(26)$$M_{V(Z)}V(Z)=Q_{V(Z)}$$
(27)$$M_{D}D=M_{U}U+M_{G}G$$
(28)$$M_{GG}G=M_{GU}U+M_{GD}D$$
where $K={\left(\frac{a}{\Delta T_{f}}\right)}^{2}\cdot \frac{h^{'}}{h}\cdot r_{o}h_{2}M$; $U,Y,V,Z,\ddot{U},Q,Q_{Y},Q_{V},Q_{Z},D,G$—vectors of additional deflections and derivatives with respect to time *t*, initial deflections, components of the stress function, geometric and material parameters, radius *ρ*, quantity *b* (*b—*length of the interval in FDM), coefficients *δ*, *γ* (differences of radial and circumferential displacements of points in the middle surfaces of facings) and number *m* of buckling waves;$P,P_{L},M_{Y}M_{V},M_{Z},M_{D},M_{G},M_{GG},M_{GD},M_{U},M_{GU}$—matrixes with elements composed of geometric and material plate parameters, the quantity *b*, radius *ρ*, the number *m*, respectively.

The system of Equations (24) – (28) was solved using the Runge–Kutta–Gill integration method for the initial state of the plate. The way to solve the problem of the static stability of a thermally loaded plate is employed in [35] in detail. The main elements of the solution process are as follows: the presentation of the stress function as the solution to the Equation (6) in the following form:(29)$$Y=-\frac{1}{2}CS\rho \ln\rho +\frac{CS\rho }{4}+e_{1}\rho +\frac{e_{2}}{\rho }$$

Assuming the following temperature conditions ${\left.T_{N}\right|}_{\rho =\rho_{i}}=T_{i}\;,\;{\left.T_{N}\right|}_{\rho =1}=T_{o}$, the form of stress function *Y* (29) is presented as follows:(30)$$Y=\Delta T_{cr}\frac{1}{\ln\rho_{i}}S\left(-\frac{1}{2}\rho \ln\rho +\frac{\rho }{4}+\rho E1+\frac{E2}{\rho }\right)-\frac{ST_{o}}{1-\nu }\rho$$

Finally, following the introduction of the dimensionless quantities and expressions of shape function and the use of the system of equations, which were established by orthogonal method for elimination of the angular variable *θ* and the finite difference method for approximation of the derivatives with respect to *ρ* by the central differences at the discrete points the problem solution comes down to the calculations of the critical temperature difference ∆*T_cr_* as the minimal value solving the eigen-value problem:(31)$$\det\left(M_{APT}-\Delta T\;M_{ACT}\right)=0$$
where ***M****_APT_, **M**_ACT_*—matrixes of elements composed of geometric and material parameters of plate, the quantity *b*, and the number *m*.

This next section may be divided by subheadings. It should provide a concise and precise description of the experimental results, their interpretation, as well as the experimental conclusions that can be drawn.

## 4. Example Analyses

The presented results will show a composite plate response on the temperature field, the action of which on the plate edges is either static or dynamic. Different combinations of laminas failures composed of fiber or matrix cracks create various symmetrical or asymmetrical plate transversal structures. The distribution of the critical temperature differences calculated between edges of plates with undamaged and damaged facings is the image of the plate reactions and shows plate sensitivity to thermal environment.

### 4.1. Plate Structure Parameters and Loading Coefficients

Accepted in numerical calculations material, geometrical and loading parameters of examined plate models are presented in Table 3. Glass/epoxy composite is an orthotropic material of the facing lamina treated as thermally isotropic or thermally orthotropic. Then, the full orthotropic properties of glass/epoxy lamina are taken into account [40,44]. The core material is made of polyurethane foam.

Using Equation (10), the engineering constants *E*, *G*, *ν* of isotropic facings were calculated for two facings examples: facings with undamaged laminas and for facings with all laminas with damages in the form of fiber cracks.

Thermal environment is characterized by an axisymmetric flat temperature field. Temperature fields, which surround the plate edges, cause a positive thermal gradient when inner temperature *T_i_* is higher than outer temperature *T_o_* (*T_i_* > *T_o_*) and negative (*T_i_* < *T_o_*), (see, Figure 1). Two profiles of temperature field are analyzed: static with fixed calculated temperature difference between the plate edges, expressed by the critical one ∆*T_cr_* and dynamic ∆*T_crdyn_* with temperature differences increasing in time according to the linear Equation (1); and assumed growth rate *a* for a positive temperature gradient.

The fiber-reinforced laminate plate facing consists of four laminas arranged as follows: [0/−45/45/90]. The configuration and the kind of lamina damages are shown in Figure 3. The correction parameter *η* (see, (12)) in the accepted damage theory of composite facings is equal to *η* = 0.1 [21]. Figure 3b shows the examples of facing laminate structures under analysis. The following cases were taken into account: facing with all laminas undamaged, facing with damaged lamina no. 1 (see, Figure 3a), and facing with damages in all laminas in the form of a crack of fiber or matrix.

Dynamic stability exemplary analyses focus on the observation of axisymmetrical plate models, whose preliminary geometry is without circumferential waves for number *m* equal to *m* = 0.

### 4.2. Convergence Analysis for FDM Plate Model and Comparison of FDM and FEM Results

The calculations carried out using the finite difference method are proceeded by the selection of the number N of discrete points. Table 1 and Table 2 present the results of the critical static temperature ∆*T_cr_* and the dynamic temperature ∆*T_crdyn_* for an FDM plate model with different number N of discrete points equal to N = 11, 14, 17, 21, and 26. Results were calculated for plates with damaged structure in the form of all fibers cracked in both laminate facings. The differences of critical values ∆*T_cr_* and ∆*T_crdyn_* presented in Table 1 and Table 2 are very small. The number *N* = 14 of the discrete points was chosen in the FDM numerical calculations.

The comparison of values of critical temperature differences ∆*T_cr_* and ∆*T_crdyn_* obtained for the FDM and FEM models of plates with isotropic undamaged and damaged both facings is shown in Table 4. All fibers are cracked in damaged facings laminates. The plate is subjected to a positive temperature gradient. Obtained results: values and bucking modes are comparable for asymmetric plate cases. Critical static and dynamic temperature differences are smaller for the FEM plate model. Minimal critical static temperature differences ∆*T_cr_* correspond with the number *m* = 2 or *m* = 3 of circumferential buckling waves. The comparison of results calculated for the axisymmetric plates *m* = 0 shows that critical dynamic temperature differences are higher than the static ones.

Additionally, Table 5 shows a comparison of dynamic temperature differences ∆*T_crdyn_* calculated for the FDM and FEM plate models. The presented values refer to plates with an undamaged structure and symmetrically transversally damaged by cracks of all fibers in laminate facings. Then, the plate structure has the quasi-isotropic properties. Calculations were carried out for five plate modes: axisymmetric one (*m* = 0) and asymmetrical ones (*m* = 1 ÷ 4). Critical temperature differences ∆*T_crdyn_* for FEM plate model are lower than those obtained for the FDM plate model but the level of values is comparable. Both plate models show the asymmetrical (*m* ≠ 0) waved form of plate buckling as this one, whose value of temperature difference ∆*T_crdyn_* corresponds to the minimal one.

Curves of time histories of deflections, presented in Figure 2, show two groups of results of the quasi-isotropic plate FDM model with both facings undamaged and damaged in the form of all fibers cracked. Black dots express the moment of the dynamic stability loss of axisymmetrical (*m* = 0) and asymmetrical (*m* ≠ 0) plates. Detailed critical time values are presented in Table 5. Critical parameters vary in time but the range of dimensionless deflections equal to 0.2 ÷ 0.3 is similar for both groups of examined plates. Using the expression (16) to describe the dimensionless additional deflection, values of the plate critical deflection can be calculated. The values are smaller than plate total thickness, equal to 1.2 ÷ 1.8 mm.

Summarizing presented observations, it is noticeable that:temperature differences cause plate buckling in a circumferentially wavy form;the thermal environment with the temperature growth in time increases the critical value of temperature differences, which are the reason for the plate dynamic stability loss;failure of lamina structure changes the response of the plate to thermal loading. Then, plate buckling occurs for higher values of critical temperature differences;the comparison of ∆*T_cr_* and ∆*T_crdyn_* values calculated using the FDM and FEM plate models shows their comparability and a similar nature of changes;the critical deformation expressed by the values of the critical deflections does not differ for the examined plate models with health facings and fully damaged ones by fiber cracks.

### 4.3. Static Temperature Field

The effect of the static temperature field on the values of the critical temperature differences ∆*T_cr_* is shown graphically in Figure 4. The distribution of the ∆*T_cr_* values depends on the combination of the damages in laminated facings. The presented results are obtained for isotropic thermal properties of facings laminate characterized by the fixed expansion coefficient *α*. The examined FEM plate model is subjected to a positive (*T_i_* > *T_o_*) or negative (*T_i_* < *T_o_*) temperature gradient (see, Figure 4a,b, respectively). The results are presented graphically which makes it possible to observe the nature of the ∆*T_cr_* changes depending on various damages and the increasing degree of failures. The shown changes are regular in the directions indicated by lines. The minimal value of ∆*T_cr_* is for the plate with both healthy facings. Damages, which disrupt the material continuity, increase the value of ∆*T_cr_*. There are observed combinations of damages, which make the plate structure transversally asymmetrical, as is the case of the plate with one facing healthy and the other one bearing a crack of the fiber or matrix in lamina no.1. The values of critical temperature differences of such plate structures are smaller than those obtained for plates with the transversal symmetry of distributed damages. Plates subjected to a positive temperature gradient for the most of the examined case lose stability in the form of *m* = 3 waves in circumferential plate direction (see, the plate modes shown in Figure 5). Cases of m = 2 or m = 4 modes are observed, too. The buckling mode *m* = 4 exists for plates subjected to a negative τεμπερατυρΔε gradient but cases of *m* = 3 are observed, too.

The same graphical presentation of the results obtained for plate models with full mechanical and thermal orthotropic properties of the laminate material of facings is shown in Figure 6. The impact of the thermal environment on the plate buckling reaction is similar. The minimal value of ∆*T_cr_* is observed for the plate with healthy facings but the maximum one is for a plate which has been completely destroyed with its both facings’ matrices cracked. The critical temperature differences ∆*T_cr_* are higher than those for a plate model with thermal isotropy. The buckling modes are not circumferentially regular. The tendency to number *m* = 3 or *m* = 4 of waves exists (see, the plate modes shown in Figure 7).

To summarize, one can notice that:both the thermal model of temperature field and properties of facings laminate material are of significance in the process of evaluation of the plate reaction to the impact of the surrounding environment;damaged plates with a transversal symmetry of structure and a smaller failure in the form of cracks located only in the single lamina lose their stability for smaller values of temperature differences ∆*T_cr_* than other destroyed plates. Smaller ∆*T_cr_* values are observed for plates with an asymmetrical structure when one of the facings is healthy;the temperature gradient direction influences the values ∆*T_cr_*;the buckling mode is circumferentially waved;the reaction of plates with heavy failures to the temperature environment is smaller than when plates are insignificantly damaged or completely healthy. Such an observation can be helpful in the process of diagnosing laminate composite structures with different damages.

### 4.4. Dynamic Temperature Field

The model of the dynamic changes of temperature fields has been expressed by the temperature linearly increasing in time with the exemplary accepted growth equal to a = 200 K/s (see, (1)). The process of time histories of deflection and velocity of deflection for the FDM and FEM plate model with quasi-isotropic facings is shown in Figure 8. The examined plate subjected to a positive temperature gradient is axisymmetrical (*m* = 0, see, Figure 5). Isotropic temperature material properties are analyzed. According to the accepted criterion of the plate stability loss (marked point in Figure 8a), the calculated values of critical dynamic temperature differences ∆*T_crdyn_* are as follows: ∆***T_crdyn_* = 81.7 K for the FDM plate model, and ∆***T_crdyn_* = 70.82 K for the FEM plate model. The time histories obtained for the axisymmetrical *m* = 0 FEM plate model with undamaged facings, also subjected to a positive temperature gradient, are shown in Figure 9. The value of ∆*T_crdyn_* is equal to ∆***T_crdyn_* = 78.82 K. Results presented in Figure 8 and Figure 9 indicate regions of values of displacements and velocities of displacements, which should be treated as showing the critical states of plates. Calculated values of ∆*T_crdyn_* belong to these regions and identify the thermal critical state of the plates under examination.

Comparing the quasi-isotropic FDM, FEM, and FEM plates, whose fibers in all facings laminates (see, Figure 10 and Figure 11) are cracked, one can observe a similar nature of both the displacement curve growth and regions of critical values for quasi-isotropic plate models (see Figure 10). Time history of the velocity of deflection of the FEM plate model with all laminate facings damaged (see, Figure 11) indicates the time moment of the plate stability loss equal to t = 0.59 s. The values of the critical dynamic temperature differences ∆*T_crdyn_* are as follows: ∆***T_crdyn_* = 129.1 K for the quasi-isotropic FDM plate model, ∆***T_crdyn_* = 113.82 K for the quasi- isotropic FEM plate model, and ∆***T_crdyn_* = 118.82 K for FEM plate model with all fibers cracked.

The graphical image of the distribution of values of critical dynamic temperature differences ∆*T_crdyn_*, which depends on the grade of the FEM plate structure failure, is shown in Figure 12. The presented results are obtained for axisymmetrical (*m* = 0) plates with thermal isotropy and subjected to a positive temperature gradient. Calculated values increase with the increase in the laminate degradation. The changes of values ∆*T_crdyn_* are similar to those shown in the static analysis. The minimum values of ∆*T_crdyn_* correspond to the asymmetrical (*m* ≠ 0) plate mode. Then, the values of ∆*T_crdyn_* are a little smaller than those obtained for the axisymmetrical (*m* = 0) mode. Exemplary results are presented in Table 5 for the dynamically loaded plates with a symmetrical cross-section built of undamaged facings or destroyed in the form of all laminate fibers being cracked.

The failure of the matrix of laminate facings discloses an interesting very high value of ∆*T_crdyn_* (∆***T_crdyn_* > 189.92 K—see, Figure 12) and different time histories of displacement and velocity of displacement. The curves are shown in Figure 13. The loss of possibility to work of such damaged plate structure is clearly seen by very small values of displacements.

To summarize, one can formulate the following remarks:presented results show the influence of dynamic temperature model on response of the composite plate;values of dynamic temperature difference ∆*T_crdyn_* are higher than those calculated in the static analysis but correspond with the changes of the static ones and confirm the observed effect of the static temperature environment on the buckling phenomenon of plates with varying grade of failure;numerical analyses show a possibility to use the temperature field in the failure diagnostic process.

## 5. Conclusions

The presented problem complements the recognition of the behavior of the complex composite structure working in variable conditions. The object of the consideration is the three-layered annular plate with different damaged laminate facings subjected to the action of the static or dynamic temperature field model. Three aims of this contribution were formulated and realized. The main one is to evaluate the stability response of composite layered plate with damages under thermal loading. Two others are as follows: presentation of the analytical and numerical procedure of problem analysis using the mathematical calculations of structure rigidity and finite element method in plate modelling, and proposal of the analytical and numerical solution based on approximation methods including orthogonalization and finite difference to calculation of three-layered plates with laminate, quasi-isotropic, undamaged, and damaged facings. Furthermore, the results obtained for the selected FEM and FDM plate models were compared.

The undertaken analyses show the responses of an examined object, i.e., a three-layered annular plate to the action of the thermal environment. The values of the temperature differences, which cause the static or dynamic loss of the stability of plates with undamaged health of layers with defects, were evaluated in detail. Various cases of possible structure failures have been analyzed to show a system of reactions of examined plate models. The created combinations of variously located damages approximate the examined structure to the real one with the lamina defects distributed randomly and diversely in the two plate facings. An approach to create the full image of plate stability responses has been undertaken.

The presented results have both a scientific aspect and a practical one in the evaluation of the behavior of the mixed composite structure with damaged and healthy laminated layers. Basically, the values of temperature differences obtained for statically and dynamically loaded plate models increase with the growth of the lamina degradation degree. Healthy plates without any cracks of fibers or matrix are particularly subjected to the loss of stability due to the action of temperature gradient. The higher values of temperature differences indicate an existence of some structural damages. This observation can be fundamental, particularly to conduct the qualitative diagnostic evaluation of the plate structure. In the exploitation process, such structure can be subjected to mechanical loads, too. Then, the depressing of the laminate fibers or matrix with diversely situated cracks becomes highly unfavorable and dangerous. The presented analyses have important elements of scientific novelty. They open a possible area of examination of a laminated structure under variable thermo-mechanical conditions. The experimental analysis, including both temperature conditions of performance of composite structure and thermal parameters, which exist during the technological laminate hardening process, could be a very important addition to the undertaken numerical examinations. This is because a consideration of the thermal effects in the laminate stress analysis is necessary above the hardening temperature.

The presented results show the essential influence of the temperature field parameters, such as the values of the critical temperature differences and the direction of the temperature gradients on plate responses can lead to the further analyses of the effects of additional parameters of the thermal environment surrounding the plate. Such investigations will be presented in the next publication.

## Figures and Tables

**Figure 1 materials-14-04559-f001:**
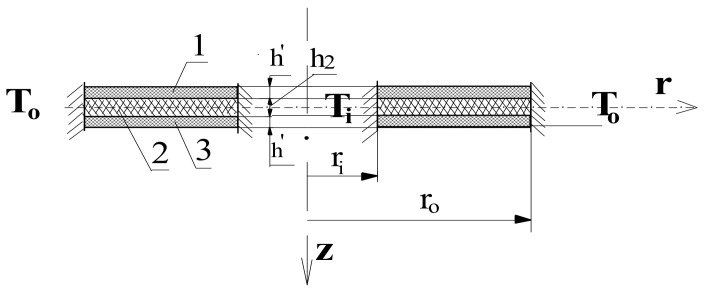
Scheme of the three-layered annular plate composed of facings (layers 1, 3) and core (layer 2) subjected to axisymmetrical temperature field expressed by *T_i_*, *T_o_*.

**Figure 2 materials-14-04559-f002:**
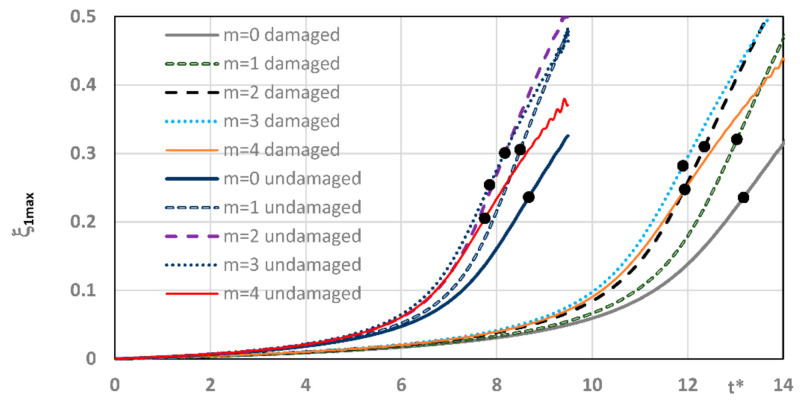
Time histories of deflection of quasi-isotropic FDM plate with health and damaged facings in the form of all cracked fibers.

**Figure 3 materials-14-04559-f003:**
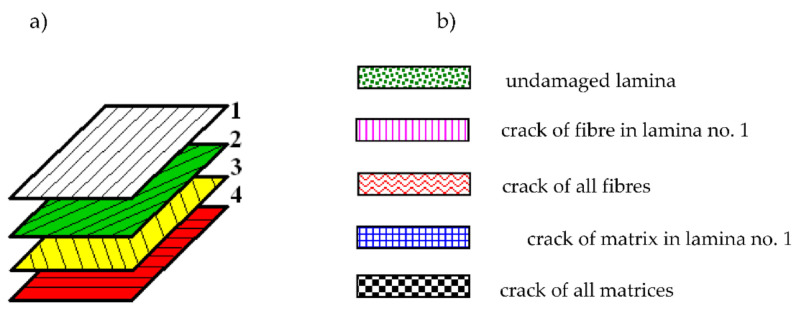
The configuration of laminate [0/−45/45/90] (**a**), the key to graphic description of laminate failure (**b**).

**Figure 4 materials-14-04559-f004:**
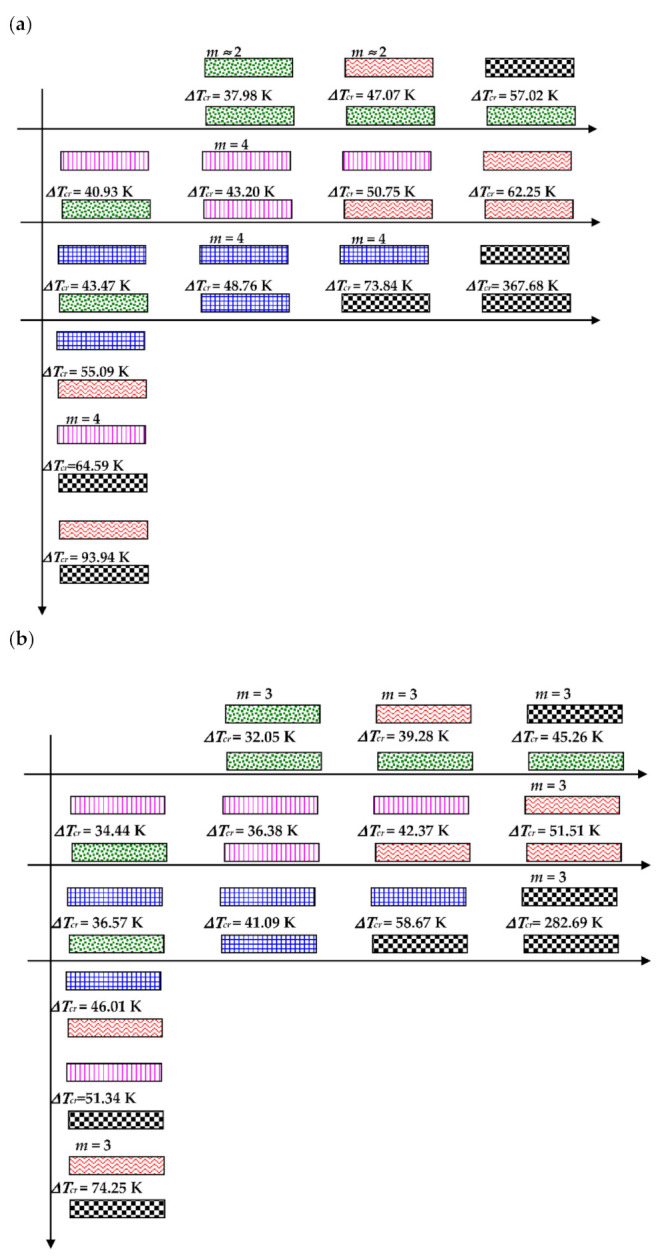
Distribution of the values of critical, temperature differences ∆*T_cr_* depending on the form of the structure failure for a thermally isotropic plate subjected to a positive **(a**), negative (**b**) temperature gradient.

**Figure 5 materials-14-04559-f005:**
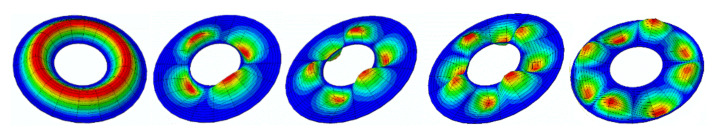
Forms of the buckling modes for a plate with thermal isotropic.

**Figure 6 materials-14-04559-f006:**
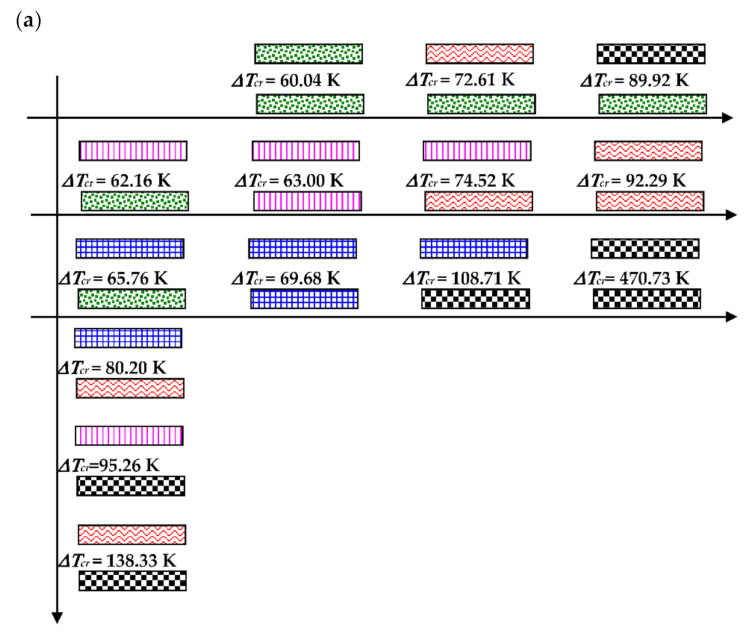
Distribution of the values of critical, temperature differences ∆*T_cr_* depending on a form of the structure failure for thermally orthotropic plate subjected to a positive (**a**), negative (**b**) temperature gradient.

**Figure 7 materials-14-04559-f007:**
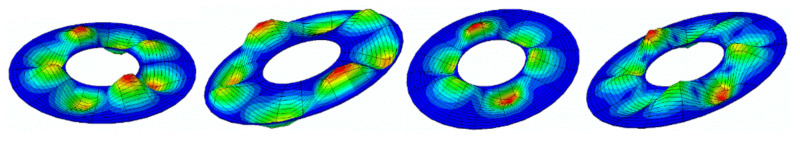
Forms of buckling modes for plate with thermal orthotropic.

**Figure 8 materials-14-04559-f008:**
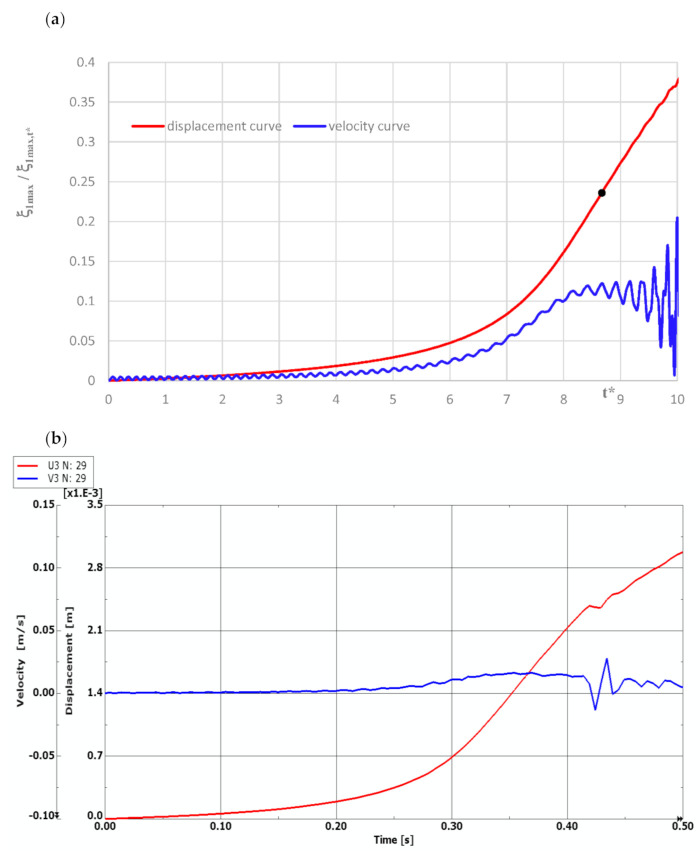
Time histories of deflection and velocity of deflection of quasi-isotropic plate with undamaged facings represented by (**a**) FDM model, (**b**) FEM model.

**Figure 9 materials-14-04559-f009:**
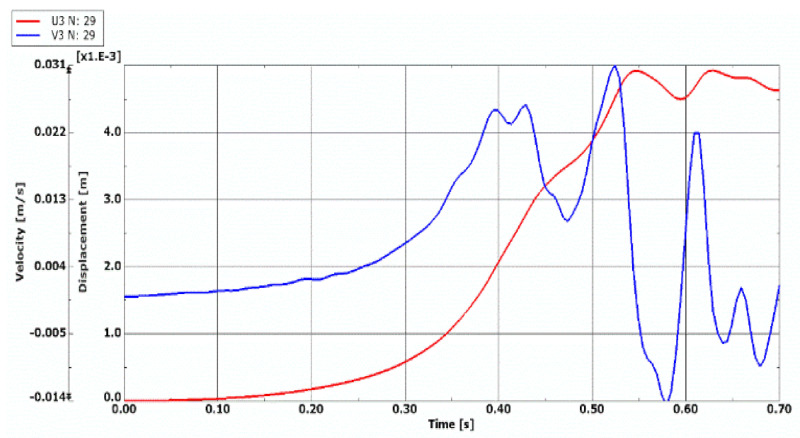
Time histories of deflection and velocity of deflection of undamaged FEM plate model.

**Figure 10 materials-14-04559-f010:**
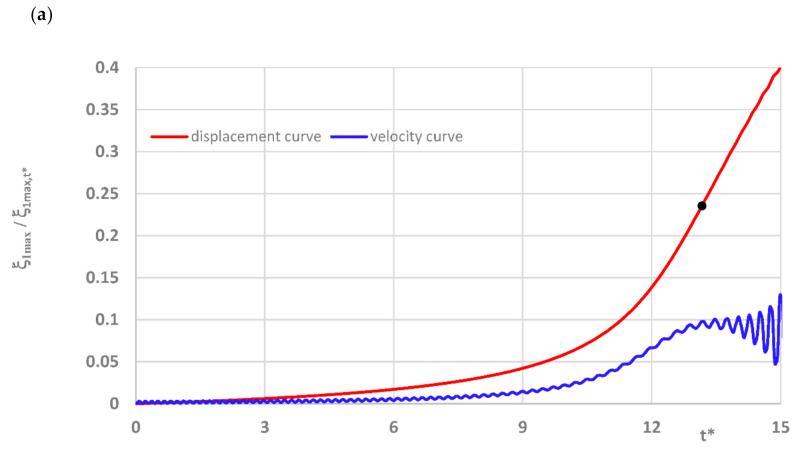
Time histories of deflection and velocity of deflection of quasi-isotropic plate with all fibers cracked in damaged facings represented by (**a**) FDM model, (**b**) FEM model.

**Figure 11 materials-14-04559-f011:**
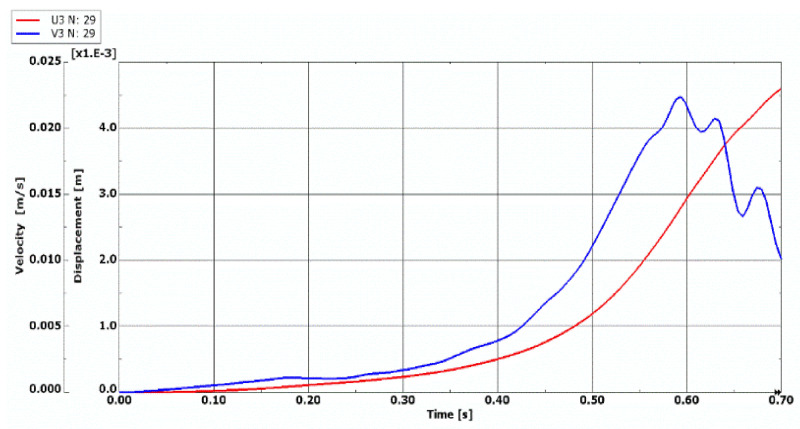
Time histories of deflection and velocity of deflection of damaged FEM plate model with all fibers cracked.

**Figure 12 materials-14-04559-f012:**
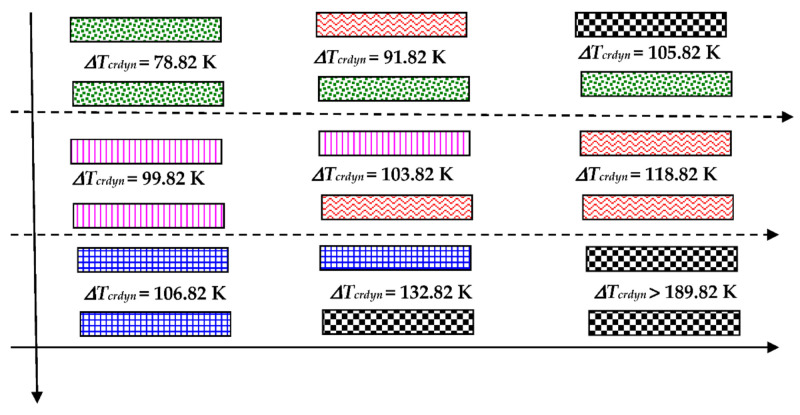
Distribution of the values of critical, dynamic temperature differences ∆*T_crdyn_* depending on the form of the Scheme 13. Time histories of deflection and velocity of deflection of the damaged FEM plate model with laminate matrices cracked.

**Figure 13 materials-14-04559-f013:**
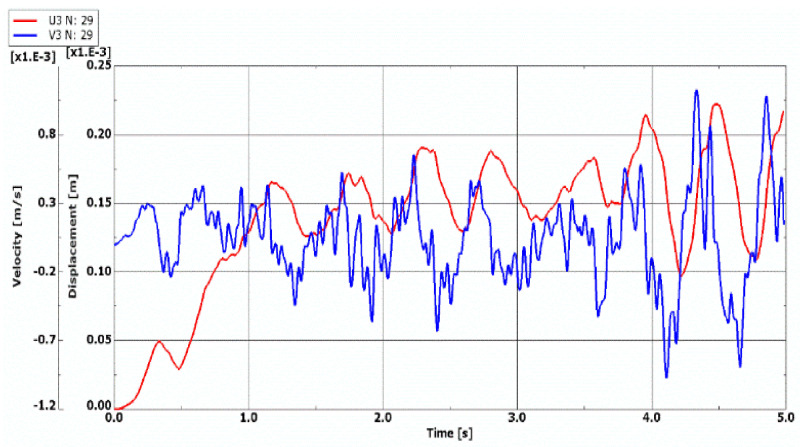
Time histories of deflection and velocity of deflection of the damaged FEM plate model with laminate matrices cracked.

**Table 1 materials-14-04559-t001:** Critical static temperature differences ∆*T_cr_* of an FDM plate model loaded thermally with a positive gradient versus a different number *N* of discrete points.

m	∆*T_cr_* [K]				
	*N* = 11	*N* = 14	*N* = 17	*N* = 21	*N* = 26
0	112.86	111.95	111.46	111.11	110.85
1	111.02	110.21	109.77	109.45	109.23
2	107.51	106.96	106.67	106.45	106.31
3	106.54	106.26	106.12	106.03	105.98
4	109.08	109.03	109.02	109.03	109.06
5	112.69	112.82	112.92	113.03	113.13

**Table 2 materials-14-04559-t002:** Critical dynamic temperature differences ∆*T_cr_* of axisymmetrical FDM plate model (*m* = 0) loaded thermally with a positive gradient versus a different number *N* of discrete points.

**Number *N***	11	14	17	21	26
∆*T_crdyn_* [**K**]	139.1	131.7	125.6	121.9	120.6

**Table 3 materials-14-04559-t003:** Material, geometrical and loading parameters of examined plate models.

Geometrical Parameters			
inner radius *r_i_*, m	0.2		
outer radius r_o_, m	0.5		
total facing thickness *h’*, mm	0.5		
thickness of each lamina *h”,* mm	0.125		
core thickness *h_2_*, m	0.005		
**Material Parameters**			
orthotropic glass/epoxy composite of facing lamina		polyurethane foam of core	
Young’s modulus *E*_1_, GPa	53.781	*E*_2_, MPa	13
Young’s modulus *E*_2_, GPa	17.927		
Kirchhoff’s modulus *G*_12_, GPa	8.964	*G*_2_, MPa	5
Poisson’s ratio *ν*_12_	0.25	ν_2_	0.3
mass density *μ*, kg/m^3^	2900	*μ*_2_, kg/m^3^	64
linear expansion coefficient *α*, 1/K for material thermally isotropic	1.96 × 10^−5^	*α*_2_, 1/K	70 × 10^−6^
linear expansion coefficient *α*_11_, 1/K linear expansion coefficient *α*_22_, 1/K for material thermally orthotropic	6.295 × 10^−6^ 20.504 × 10^−6^	-	-
calculated engineering constants of quasi-isotropic facings			
facings with undamaged laminas		facings with damaged all laminas (fiber cracks)	
Young’s modulus *E*, GPa	31.1	18.71	
Kirchhoff’s modulus *G*, GPa	12.5	7.91	
Poisson’s ratio *ν*	0.245	0.182	
**loading parameters**			
rate of thermal loading growth *a*, K/s	200		
fixed temperature difference ∆*T*_f_, K	10		

**Table 4 materials-14-04559-t004:** Critical static and dynamic temperature differences of the FDM and FEM plate models loaded thermally with positive gradient.

Plate Model	Asymmetric Plate *m* ≠ 0	Axisymmetric Plate *m* = 0	
	Static Analysis	Static Analysis	Dynamic Analysis
	∆*T_cr_* [K]	∆*T_cr_* [K]	∆*T_crdyn_* [K]
	Undamaged Facings		
FDM	69.36/3	72.14	86.70
FEM	62.20/2	64.35	70.82
	**Damaged Facings**		
FDM	106.26/3	111.95	131.70
FEM	94.05/3	98.06	113.82

**Table 5 materials-14-04559-t005:** Critical dynamic temperature differences of the FDM and FEM plate models with different modes loaded thermally with positive gradient.

Plate Mode	∆*T_crdyn_* [K]			
	Undamaged Facings		Damaged Facings	
	FDM Model	FEM Model	FDM Model	FEM Model
0	86.7	70.82	131.7	113.82
1	84.9	67.82	130.3	99.82
2	81.7	67.82	123.5	103.82
3	78.5	66.82	119.0	105.82
4	77.5	70.82	119.4	109.82

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
