# Peer review of "Temperature Effect on Stability of Clamped–Clamped Composite Annular Plate with Damages"

_materials, 2021, doi:10.3390/ma14164559_

Round 1

Reviewer 1 Report

This work presents interesting study to examine laminated structure under variable thermo-mechanical conditions.  Both analytical and numerical methods were employs to study the laminate structure stability, and in my opinion the results do quite meet the criteria of this journal. The impact of this study on composite community would be moderate to high, therefore, I recommend accepting this paper for publication.

Author Response

I would like to kindly thank for review of manuscript and very nice opinion.

According to suggestions presented by the Reviewers 2 and 3 the manuscript has been improved. I would like to present its new version. The new fragments are presented in red colour. The new two Tables with data and notations have been added.

Reviewer 2 Report

  1. The current study investigates the performance of a three layered plate that contains different damage facings due to the action of the static/dynamic temperature field model. The author study how the plate stability is affected due to temperature experimentally and numerically. The authors apply two FE methods, FEA and FD along with an analytical model which takes into account the temperature effects and clamping conditions. The author found that the response of the plate depends on the amount of damage in the laminate elements.  
  2. The abstract does not read well, first of all, the results are somewhat vague and do not show fundamental findings from the study. Why the author conducted the FE study, is it just because they can? What is the aim of using FD and FEA methods?
  3. Please consider reviewing the abstract and highlight the novelty, major findings and conclusions.
  4. The authors sometimes write small paragraphs of 1-3 lines, this is not encourage as it breaks the flow and readability of the manuscript. The author should combine any small paragraphs into larger ones. For example, 141-142 please combine it with the previous or afterward paragraph. Please check for this issue everywhere else in the manuscript.
  5. The author need to follow the guide for authors and use the template of the journal properly. First of all, at the end of the introduction the authors should answer the following question: What is the research gap did you find from the previous researchers in your field? Mention it properly. It will improve the strength of the article.
  6. Also after answering the previous question, the authors should provide a brief summary of what was done in this paper, this is a standard format in any scientific article.
  7. The author should add a list of nomenclature at the end of the manuscript for all the symbols and Greek letters reported in this work.
  8. Problem Solution this is a very long section, perhaps the author should consider moving most of it to an abstract and only keep the main formulas and final derivations. For example, why the author need to give us 101 revision of classical plate theory?
  9. The authors copy way too much information of basic plates theory in their article which is not acceptable, after reading it further, the author must move all this section into an appendix, it is not acceptable this way.
  10. Example Analyses perhaps change this name to analytical model.
  11. The author is encouraged to add all the FEA and FD data (material properties) into tables rather than listing them in a paragraph as they are so many.
  12. The author uses different parameters from different sources and use them all in their model, are all the data used belongs to sample material?
  13. Please mention some of the limitations in your FEA and FD models.
  14. The author need to restructure and organise the manuscript properly and not as a report or a chapter in a thesis. Please check the guidelines of the journal.
  15. Figure 5 and 7 are pointless, they do not even have the coloured legend and what it represents.
  16. Figure 8 b and figure 9 are very difficult to read (not clear). Figures 10b and 11 same.
  17. The results are merely described and is limited to comparing the experimental observation. The authors are encouraged to include a discussion section and critically discuss the observations from this investigation with existing literature.

Author Response

I would like to kindly thank for review of manuscript and all presented remarks, which are very valuable.

Most of suggested elements have been improved. All changes in text are marked in red colour. The data and main notations are presented in additional Tables.

Abstract has been changed. The new fragments, which taken into account on the aims of the contribution, realised elements and connected with them achievements have been added.

Figures can be only increased. Figures of plate modes show only the kind of the form of buckling, so the legend is not necessary.

The fragments with analytical solution have not been changed. There are selected relations, which are in my opinion important for presented paper.

Reviewer 3 Report

The paper presents the response of a three-layered annular plate with different damaged laminate facings to the action of the static or dynamic temperature field model. The problem of the plate stability loss in the temperature environment has been solved analytically and numerically. Obtained results indicate practical meaning of analyses in failure diagnostic process.

Generally speaking, the manuscript is well written, the material is judiciously divided and organized and correct from scientific point of view. Some changes are, however, necessary. For these reasons I can recommend the acceptance of this paper after some corrections.

Before that the Editor makes a decision I suggest that the authors emphasize take into account the following corrections

1     The section Conclusions will be point out the original results of the paper and can be extended to highlight the contributions.

  1.     I think the authors need to emphasize more clearly the contribution of the manuscript from a scientific point of view.

If the author takes into account these observations the work can be published.

Author Response

I would like to kindly thank for review of manuscript and all presented remarks, which are very valuable.

The paper according to the suggestions has been improved. The new fragments, which taken into account on the aims of the contribution, realised elements and connected with them achievements have been added. The scientific worth of work has been underlined.

The changes are presented in red colour.

Round 2

Reviewer 2 Report

All questions answered and paper can be accepted